# Constructing the Transitions and Co-Existence of Rural Development Models

**Marcelo Sili** [1,*], **María Isabel Haag** [2] **and María Belén Nieto** [2]

1    National Scientific and Technical Research Council (CONICET), Universidad Nacional del Sur, Bahía Blanca 8000, Argentina

2    Departamento de Geografía y Turismo, Universidad Nacional del Sur, Bahía Blanca 8000, Argentina; isabel.haag@uns.edu.ar (M.I.H.); mbnieto@uns.edu.ar (M.B.N.)

*    Correspondence: sili@uns.edu.ar

**Abstract:** The rural world in Latin America is becoming more complex and diverse. In recent decades, new non-traditional productive activities have emerged, technological change has increased, local culture and tradition have been revalued and rural tourism has been developed, among other dynamics. Scientific and technological change, greater concern for the environment and new consumption patterns are at the root of these changes. It can be said that Latin America has begun a process of transition to new models of rural organization and development. Returning to the conceptual framework of innovation, the hypothesis underlying this work is that emerging initiatives constitute niche activities which, over time, become integrated into the territories, resulting in two types of situations: co-presence of activities and actors, with conflicts and competencies that prevent the construction of synergistic development dynamics, or co-existence, with shared articulations and projects between activities and actors. To account for this hypothesis, three experiences in Argentina are analyzed: one is the emergence of agroecological activities in regions dominated by production systems linked to exports; the second case of analysis is the emergence of tourism in traditional rural areas; the third, the emergence of more innovative and sustainable livestock farming. This research is qualitative and exploratory, based on interviews with key actors in all these activities.

**Keywords:** innovation; transition; rural areas; Argentina; co-existence; co-presence

## 1. Introduction

The rural areas of Latin America have, broadly speaking, been characterised historically by the presence of two situations: on the one hand, the dynamics of productive growth in all sectors of primary production, based on the view that agriculture should be boosted as a strategy to guarantee food supply and generate foreign currency through exports, thus enabling national economies to be financed [1]. This orientation, centred on agricultural development, has clearly structured policies for much of the last century and has been responsible for the modernisation process that has resulted in strong growth in the production of goods and services based on natural resources. In many cases, this vision has been clearly criticised from different professional and political spheres for not contributing to sustaining the roots of the population, nor did it achieve broader goals, such as social and territorial balance and the environmental sustainability of rural areas. From this perspective, many regions have been transformed into productive or export platforms in recent decades, where priority is given to agricultural production and the generation of foreign currency, so that, as Mackay and Perkins state, these territories

"exist(s) only to be exploited in the pursuit of all-out profit". [1] (2019:3)

On the other hand, and parallel to this productive dynamic, there are still vast areas with a strong rural population, and family and peasant agriculture that are unable to overcome poverty, and which face structural problems, such as lack of infrastructure,

and land and water for production, where situations of environmental deterioration and increased poverty are consolidated. Policies to support these sectors have been diverse and anarchic, without sufficient strength and capacity to structure sustainable changes in rural areas and ensure the overcoming of rural poverty [2].

Beyond this dichotomous model, between the hegemonic activities that structure the vast rural spaces (forestry, soya, livestock, among others), and the persistence of peasant and indigenous family farming, new productive activities and new production models have emerged under different logics and productive scales over the last two decades. Technological change and the development of new bio-economic activities (biofuels, circular economy, medicines, new foods, among others) are being promoted, and there is a revaluation of typical productions, culture and local tradition, together with new efforts in rural tourism, among others [3–5]. A new representation and conceptualisation of "rural" is also progressively emerging (largely derived from a redefinition of its links with "urban"), whereby it is no longer seen as a space whose exclusive function is the production of food and primary goods, but is attributed new and multiple functions [6–8]. These changes are resulting in a significant transformation in rural areas.

All these changes have been driven by several factors: (a) modernisation in transport and communications, especially mobile telephony and the internet; (b) growing concern for the environment and habitat; (c) the need to protect the environment; (d) the need for a more efficient use of natural resources [9]; (e) the emergence of new consumption patterns much more oriented towards quality, healthy agri-food goods with a territorial identity; and (f), finally, a new revolution in science and technology linked to the processing of natural resources (new knowledge and technologies).

It can thus be said that rural areas in Latin America have begun a process of transition towards new models of rural organisation and development [6,10–12] with new scenarios of coexistence, co-presence and hybridisation of models and actors. In the face of these transformation processes, several key questions arise that we would like to answer: how are these innovation processes integrated into the current productive regime and into rural areas; what are the conditions for these innovations to become durably embedded in the territories and to move from the niche stage to become part of a new socio-technical regime, overcoming the existing dichotomous model; and finally, what conflicts and synergies are generated between existing activities and new initiatives?

Following the conceptual proposals of the MLP (multilevel perspective) regarding innovation [13], the hypothesis put forward in this paper is that the initiatives emerging in recent years in many rural areas of Argentina are initially constituted as products or activities that can be called niche products or activities [13–16], with a strong local specificity, but which over time and under certain conditions, become consolidated and integrated into the traditional production systems of each territory, creating new socio-technical regimes or socio-productive scenarios [17]. This passage from one regime to another would be marked by two moments or scenarios. The first scenario is the co-presence of activities and actors [18]: this is a moment in which various actors and activities are deployed in the same region, but they do not coexist or articulate with each other, but often enter into conflict or competition for the region, causing blockages, tensions, and even mutual ignorance between actors and activities (for example, agroecological production vs. soya production with high levels of chemical inputs) [19], exacerbating conditions of socio-territorial fragmentation [20], and which does not allow for the construction of more synergetic and consensual territorial projects. A second moment or scenario is called co-existence: i.e., the activities and actors (traditional and new) complement and articulate themselves in terms of possible joint projects (food production with high added value and the development of rural tourism); in these cases, the construction of a more synergetic and consensual territorial project is much more feasible and viable [21].

The consolidation of innovation processes and therefore the constitution of a new socio-technical regime would require, in principle, ensuring the passage from a scenario of co-presence of activities to scenarios of co-existence.

The aim of this paper is to observe how these dynamics of innovation and transition towards new models of development are produced, highlighting the existence of two very different logics of co-presence or co-existence. The understanding of these transition processes and the logics that govern them are key elements for the design of rural development policies in Argentina, and also in other countries where rural change processes are taking place.

This paper is based on the analysis of three experiences in Argentina, considered as a transition towards scenarios of greater sustainability of productive systems. The first is the emergence of agroecological activities in a territory dominated by traditional or conventional production systems, linked to the export of agricultural and livestock goods [22]. The second case of analysis is the emergence of tourism in rural areas. The third case of analysis is the emergence of new forms of livestock production. The first two experiences of analysis are strongly advocated by public action (municipal governments, INTA—National Institute of Agricultural Technology, universities) as activities with a strong capacity to create and structure changes in the current rural production regimes, and constitute a "promise" of transition and new modes of rural development [23]. On the other hand, the third case appears as more invisible experiences and less recovered by public action.

This paper is based on the analysis of a conceptual framework centred on the concept of innovation and socio-technical regime, and also discusses the concepts of co-presence and co-existence of development models [18,21,24–26]. Secondly, the analysis scenario and methodology are presented, followed by the case studies in the third part, and finally a discussion of the results in the light of the theoretical and conceptual contributions.

## 2. Innovation, Co-Existence and Co-Presence of a Socio-Technical System

### 2.1. Innovation and Socio-Technical Regime

Innovation can be defined as the process of creating a new product, process, service or management model that can solve a problem, increase efficiency, or open new paths or alternative solutions to complex problems.

Traditional approaches to innovation have focused on the incorporation of new technologies and the introduction of radical changes in the production system, in which the entrepreneur seeks to maximise profits [27,28]. The focus is on innovation that is produced incrementally within productive systems, in a logic of creative destruction, which assumes that in order to create something new it is necessary to totally or partially destroy the old. In general, innovation is carried out on existing resources and practices and can occur in different fields of economic activity, whether in the introduction of a new product, a new production process, the opening of a new market, the possibility of incorporating a new raw material, or the introduction of other modes of management.

Although Schumpeter focuses on individual behaviour, he also recognises the importance of the social environment in innovation. For the author, the existence of a favourable economic, cultural, political and social context can stimulate innovative behaviour [29]. This perspective has opened up new approaches to the analysis of innovation, which go beyond the technical field and individual action. The new approaches consider innovation as a systemic process, inserted in a historical, social and territorial context that conditions it (path dependency). Thus, innovation does not constitute an isolated act, the result of an individual action, but of a process of social interaction that extends over time and space in such a way that it only entails a total break with previous trajectories on rare occasions.

Innovation arises from an interactive process that brings together the knowledge and skills of many to solve common problems [28,30]. This condition underlines, on the one hand, the importance of a social fabric in which links of cooperation and exchange can be established, and on the other hand, the centrality of the territory as a sphere of proximity in which these networks of social interaction are produced and reproduced over time. Understood from a broad perspective, and conceived in terms of rural territories, innovation is no longer limited to strictly technological and economic processes, but rather

crosses many other spheres of community life, materialising in a variety of projects that include environmental protection, landscape preservation, recovery and enhancement of heritage, reconstruction of the social fabric, socio-territorial animation in depressed areas, and others [28,31].

In order to account for innovation processes and their contribution to production system change (and thus territorial change), in this paper we follow Geels' multi-level perspective [32] which considers that innovation and change processes should be viewed in terms of three levels of analysis:

(a)  The emergence of novel activities or processes that may initially be considered as niche activities, but which represent an embryo of new development models [33].

(b)  The functioning of a socio-technical regime. The concept of a socio-technical regime provides the frame of reference for understanding the functioning of the current system or model of organisation and production in the rural world [34,35]. The socio-technical regime refers to the various elements that constitute a mode of production, such as existing technologies, production and organisational practices, regulations and norms, forms of governance, infrastructures, conditions of resource organisation (land, property, spatial organisation) and socio-cultural discourses that sustain the production regime. A socio-technical regime becomes stable but changes as new innovations or disruptive processes emerge that systemically transform it. In this way, innovations should not only be seen as a novel element, capable of generating jobs, or improving the conditions of governance or environmental sustainability, but should also be valued in their capacity to modify the productive and social organisation regime of a territory, thus contributing to generate profound changes in its dynamics, and therefore in the dynamics of a territory.

(c)  The general contextual conditions in which these actions take place. This is the territorial, political, economic and cultural context of a province or country. These conditions are the ones that put pressure on the existing regime for its transformation (e.g., cultural change around the environment, new demands for food products, among others), but these conditions are also transformed by the regime itself, so that territories change according to the transformation of the socio-technical regime, or the macro-economic conditions of a country.

Figure 1, inspired by Geels' work, presents this model [15,19,36,37]. Innovative processes or niches can be recognised as a specific phenomenon, as a novelty or special case that stands out in relation to the rest of the activities in the area. Depending on the conditions of the socio-economic, political and institutional context, these innovative processes can be integrated into the dominant production systems of a territory (current socio-technical regime) according to the ideology or the expectations and visions of development held by local actors. This dynamic transforms the current regime, reconfiguring the conditions of the socio-territorial context, i.e., the forms of organisation and the dynamics of societies and, in the case we are interested in studying in this paper, of rural areas.

Applying this approach in very general terms to the Argentine case, it can be observed that the socio-technical regime of the period of agrarian modernisation between 1960 and 1980 was characterised by the development of mechanised agricultural activities, with the use of new hybrid varieties, and with conventional work and farming practices, all under the protection of technology transfer policies from public bodies and private companies. From a cultural point of view, this socio-technical regime was sustained by the idea of urban modernity, which led to the abandonment of the countryside and the concentration of the population in towns and small cities, supported by the strong and massive use of automobiles, thus generating a specific socio-territorial model [35]. In the 1990s, another socio-technical regime was constructed thanks to the emergence of new innovative processes, such as the use of glyphosate-resistant soybeans, no-till farming and the formation of investor networks [36].

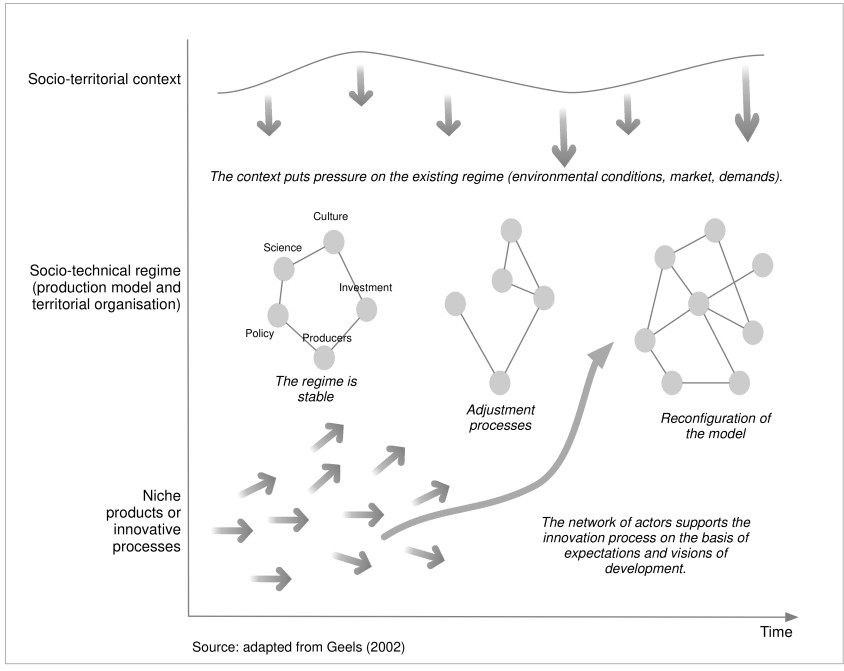

**Figure 1.** Dynamics of innovation and regime change and socio-territorial context.

This general model proposed by Geels is extremely useful for analysing innovation processes and their historical evolution [19,37,38] (see Figure 2).

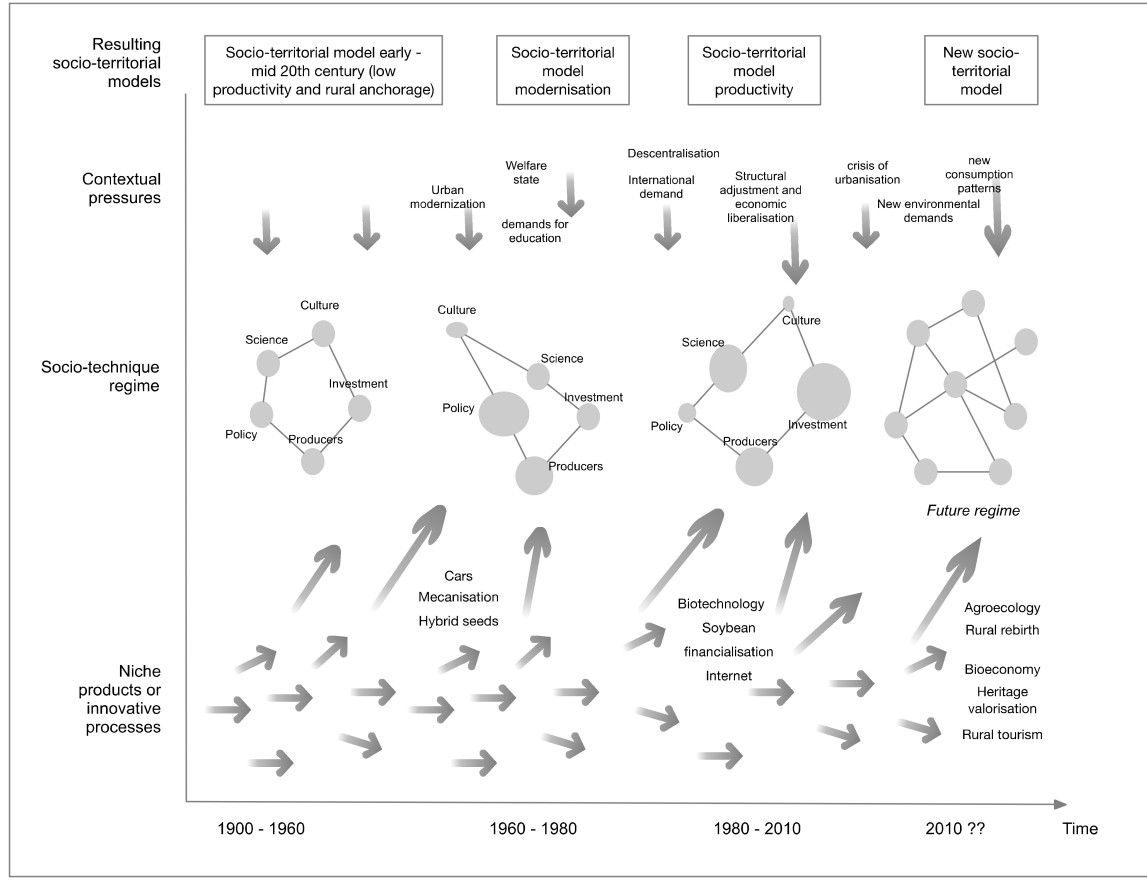

**Figure 2.** Transition of socio-technical regimes and socio-territorial models in Argentina. Source: Own elaboration based on Geels 2003.

These innovations, which were initially considered niche innovations, were gradually consolidated, substantially transforming the socio-technical regime of agrarian modernisation into a new regime characterised by the presence of global service and biotechnology companies, with more complex production logics from a technical and financial point of view, with the predominance of soya production and derivatives, and with a new articulation with the regional cities, which now played an organising role in the new regime and in the accumulation of agrarian rents.

Each of the emerging socio-technical regimes is characterised by several features [18]:

1.  A minimum scale that allows it to maintain its own dynamics and coherence and gives it visibility and recognition by other actors.
2.  An identity and its own discourse on its organisation and development. This discourse can be disseminated by the actors involved (producers, technicians, civil servants, support organisations), and allows it to be recognised by society and consumers.
3.  A general support policy that translates into bureaucratic and administrative structures, regulations, specific public policies based on the recognition that this socio-technical system generates a real contribution to the economic and productive life of the region.
4.  A form of knowledge construction and its own technical organisation, capable of sustaining productive dynamics. This organisation may be very diverse: it can be monopolised by public agencies, such as INTA, as happened in the process of agrarian modernisation, or it can be made up of networks of private companies, as happened with the socio-technical regime in the Pampa in recent decades.
5.  Specific marketing logics

In short, this whole process of change can be defined as a process of transition of models or socio-technical regimes, in which the transition is defined as follows "as a shift from one socio-technical regime to another, which, according to the multilevel perspective, happen through a combination of macro landscape pressures and micro niche developments" [37] (p. 1).

*2.2. Co-Presence and Co-Existence of Socio-Technical Regimes*

As pointed out by Truffer [38] and Gasselin [32], this process of transition or change in socio-technical regimes is neither linear nor free of conflicts and contradictions; on the contrary, it is highly dependent on other factors, such as the ways in which the different levels are articulated (local, regional, national and international) on the structural conditions of the territory, clearly pointed out by Murphy [39], and especially of the factors and logics of power involved in these processes (governance), as Konefal states [40] in his analysis of the processes of change in US agriculture. A closer reading of the processes of transition or regime change allows us to observe two very different evolutionary scenarios:

The first possible scenario is the co-presence of regimes. This implies that two or more regimes [18], embodied by various actors and activities, with their own institutions, networks and productive logics, are present in the area, but do not have any functional or synergetic articulations between them; on the contrary, models and activities can be juxtaposed, and actors can deny or ignore each other, or there may be physical or verbal violence, intimidation, domination or manipulation. In these cases, emerging innovation processes (e.g., agro-ecology, rural tourism, introduction of new technologies) are present in the area, but are not integrated into the dominant socio-technical regime, nor do they have the capacity to structure a new hegemonic socio-technical regime. This scenario is very evident in many rural regions of Latin America, where, for example, innovation processes in family farming are confronted with the more hegemonic model of highly technological corporate agriculture, creating a scenario of competition and conflict over resources.

A second situation has been called co-existence, which implies a consensus and acceptance of the existence of multiple regimes, embodied by different actors and activities, and the certain possibility of constructing a new socio-technical regime, overcoming the previous model. In this case, these actors and activities can confront each other while

maintaining their identities and projects. In other words, not only can there be functional relations between multiple actors (as in the case of industrial districts or milieux innovateurs), but there is also a process of building a shared identity. These situations are evident in regions where the hegemonic or predominant activity incorporates other actors or activities in a complementary way, generating important synergies, for example in wine-growing areas where new rural tourism activities are developed, or new fruit or vegetable crops that take advantage of the dynamics of local employment, or the infrastructures already available, creating a more diverse and complex network in which wine-growing ceases to be dominant.

The organisational capacities of the territory, its development itineraries and its capacity to build new socio-technical regimes are not the same under conditions of co-presence or co-existence [18]. In the case of the co-presence of models, it is likely that scenarios of greater conflicts, blockages, or loss of opportunities for the development of the territory will be generated; in the case of scenarios of co-existence, synergy and collaboration between actors and activities can result in conditions for building new socio-technical regimes and more solid development itineraries. Thus, innovation is not only limited to the creation of new activities and products, but also plays a key role in the construction of social and institutional governance dynamics that allow the passage from situations of co-presence to scenarios of co-existence [21,26]. Indeed, to ensure the transition from situations of co-presence (often conflictive) to situations of co-existence that are more synergic and contributive to a dynamic of territorial development, new innovations in terms of governance and territorial management (new forms of management of activities, of articulation and organisation of institutions, and of local regulations) are also required [41]. Innovation, understood in a broad sense, not only accompanies the emergence of new niche activities, but also plays a key role in their consolidation and the construction of a new stable rural development regime.

### 2.3. The Governance of the Passage from Co-Presence to Co-Existence of Socio-Technical Regimes

Governance is a polysemic concept for which there is no single definition, but which varies according to contexts and fields of application. However, governance can be defined as a system of norms, rules and practices that make it possible to organise and articulate public, collective and private action with a view to building a concerted project for the future [42–44]. It is through this form of governance that multiple actors at different levels (local, regional, national and international) can articulate and resolve the conflicts that limit the action of each of them.

Taking this definition into account, we consider that the construction of adequate governance processes and the articulation of actors is key in the transition from a scenario of co-presence to a scenario of co-existence, as it implies generating the social and political-institutional conditions to harmonise the visions, demands and initiatives of the multiple actors involved in the territory. However, as [33,40] point out, one of the main criticisms of multilevel perspectives is, in short, the lack of consideration of these political processes and the power games that take place between the three levels of analysis (landscape, regime and niches), and within the different socio-technical regimes. The criticism is centred on the lack of emphasis placed on the analysis of transition governance, on the ways in which conflicts between the multiple actors involved in the three levels of the model (innovative activities, socio-technical regime and socio-territorial context) are regulated or managed. In order to overcome these limitations, we will place emphasis on understanding the power relations between the actors present in the territory, also considering the processes of co-presence and co-existence as part of a transition process [32].

This analysis of how actors interact and mobilise at different levels according to their networks of cooperation and solidarity or their interests has become a key element in recent decades, given that these forms have changed substantially in the last half century, since the hierarchical position that the state used to occupy in development processes (being the one that planned, developed infrastructures, organised land use and defined

development models in general) has shifted to a relational power centred on networks of actors that intervene in multi-sectoral areas and at multiple levels of scale (regional, national and international), acting on diverse social fabrics and organisations that also have an increasing weight in the organisation of the territories [45,46]. Thus, it is not only public action that prevails in the construction of innovation and development dynamics, but also private and collective actors with multiple logics, very different from each other, with different times and different needs [47–49]. The logics of each type of actor are not arbitrary but respond to cultural and historical contexts that influence their actions. The interaction between actors is regulated by explicit rules that are, in general, legally formalised and institutionalised in regulatory structures, or they may be informal norms shared by the members of an organisation and implicitly governing behaviour in society. Among the formal rules, the first is the constitutional framework, which contains the constitutive rules of a state and applies to all decisions, individual freedom or political authorities. Secondly, the institutional rules that regulate the administrative organisations of the state, referring to organisational and procedural aspects. On the other hand, the informal rules circumscribing the system of values, symbols and behavioural norms specific to each society provide a framework of meaning that guides actions. This system of rules, both formal and informal, materialises the power relations between social groups, facilitating or limiting the dynamics of action. The analysis of the actors' game should include the institutional dimension that frames it, identifying the rules involved, their stability over time, the negotiation processes involved, the possible conflicts between them and their influence on the participants' behaviour.

These rules, both formal and informal, should not only be seen as regulatory frameworks for social functioning and actors, as they impose conditions, but also as frames of reference for the construction of new cultural modes, as they generate meanings about reality and cultural elements that condition the interaction between the members of an organisation.

### 3. Applying the Framework: The Case of Rural Tourism, Agroecology and Livestock in Argentina

In order to be able to account for the working hypothesis, three innovation processes were analysed that allow us to think about the transition of socio-technical regimes in Argentina, in which processes of co-presence and co-existence of socio-technical regimes are also verified. A first case of innovation corresponds to the emergence of innovative processes and rural tourism activities, the second case corresponds to the emergence of agro-ecology, and the third case to new forms of livestock production. All three cases are situated in a rural context characterised by intensive and extensive crop production and conventional livestock farming.

The study area is the District of Villarino (see Figure 3), of 1.1 million ha, which is located in the southwest of Buenos Aires province, in which it is the second largest district. According to the 2001 population census of the Instituto de Estadísticas y Censos de la Argentina, the area has a little more than 30,000 inhabitants distributed in several localities, such as Pedro Luro (9500 inhabitants), Mayor Buratovich (5600 inhabitants), Hilario Ascasubi (3500 inhabitants), Médanos (6000 inhabitants), Algarrobo (5500 inhabitants), and in the rural areas, although it is estimated that there are currently 20% more inhabitants than in the 2010 census. The axes of the national routes N° 3 and 22 structure this territory, together with the Colorado river, one of the main rivers in the region, which is used to provide irrigation to much of this vast area. The land is of flat or undulating relief, with a semi-arid climate. Rainfall averages 500 mm per year, with strong variations from year to year, which determines shrub and grassland vegetation adapted to seasonal drought conditions. Faced with these environmental conditions, the production systems are organised in two main zones.

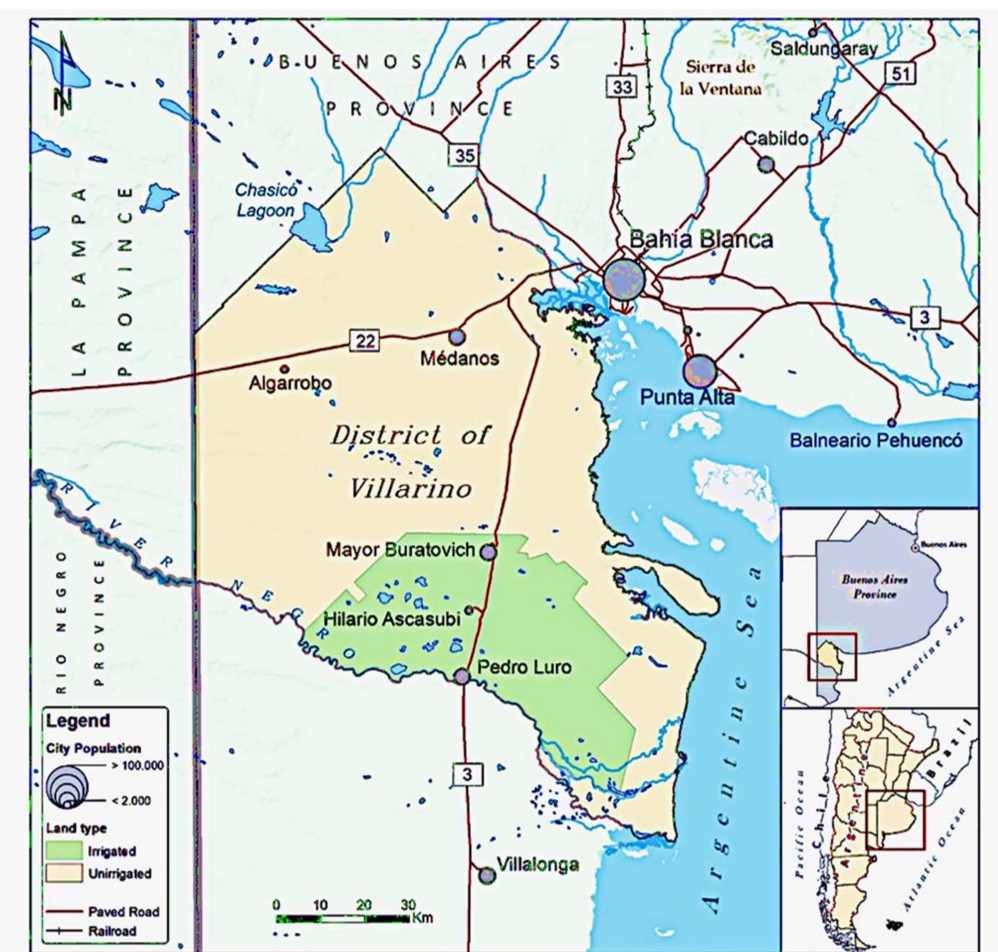

**Figure 3.** Location of the District of Villarino.

The dry land or northern zone is organised around the localities of Argerich, Médanos, Algarrobo and Teniente Origone. This is a semi-arid zone, with natural fields suitable for livestock and crop production (wheat and oats). The most characteristic production systems are mixed, with cropping and livestock or pure livestock. The surface area of livestock farms towards the west of the area (the more arid zone) is around 1000 ha, while towards the east the average surface area is 100 ha in farms that were historically dedicated to garlic production.

In the southern zone, due to the availability of irrigation water from the Colorado river, there is a wide range of possible crops and activities, with intensive and extensive crops such as sunflower, corn, sorghum, seed crops (sunflower, alfalfa), and horticulture, especially onions that are export-oriented and heavily dependent on the Brazilian market. There is an important development of livestock and dairy farming, as well as beekeeping, which is widely recognised in the region. This southern area has historically been characterised as a receiving area for European immigrants at the beginning of the 20th century, and immigrants from neighbouring countries since the 1970s and 1980s, who arrived as seasonal workers in the horticultural activities, and later settled in a stable way. The type of productive activity based on irrigation and intensive land use has defined much smaller farms than in the rainfed areas.

This territory has historically had different socio-technical regimes that configured specific forms of organisation for each period, although these regimes are very different, as the northern zone is without irrigation and the southern zone has irrigation. Figure 4 presents these different production regimes.

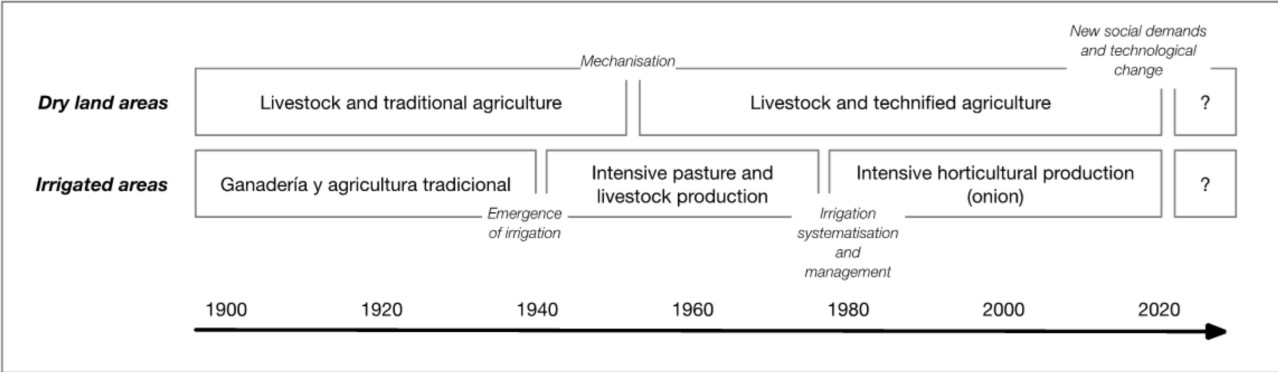

**Figure 4.** Socio-technical regimes in the District of Villarino, Source: own elaboration.

In the rainfed areas (without irrigation), the regime was characterised by traditional livestock and crop production, with the greatest processes of innovation and change occurring during the 1960s with mechanisation and the adoption of new technologies.

In the irrigated areas, the land was organised at the beginning of the century, as in the northern zone, with livestock and, to a lesser extent, cropping activities. The most profound changes took place with the incorporation of irrigation for pasture production, and with the process of systematisation and organisation of irrigation, a new system was consolidated around onion production.

Both areas, north and south, are now facing new changes and challenges. The main problem is the lack of water, the result of several years of drought (due to low rainfall) and low snowfall in the Andes mountains (the main source of irrigation water for the Colorado river), which has substantially reduced the irrigated area from 140,000 ha to approximately 30,000 ha that are irrigated effectively today. Other problems have been soil and environmental degradation, and changing market conditions. Faced with these problems, landowners in the irrigated areas have begun to choose other types of production that are more resistant to salinity and lack of water, favouring the production of fodder crops and abandoning traditional onion production, an activity that has been moving to the Negro River valley.

In short, these problems are leading many small and medium-sized producers to look for new productive alternatives outside the highly water-dependent cropping activities (onions and cereals), or to make changes in the management and administration of their activities. Within these innovative processes, we can find new agro-ecological practices and rural tourism, activities that are strongly supported by INTA (Instituto Nacional de Tecnología Agraria), the Cambio Rural (rural change) programme (Ministerio de Agricultura), the ProHuerta programme (Ministerio de Desarrollo Social), and the universities in the neighbourhood. These processes are supported as the new "promises" to cushion the current production crisis and to build a new socio-technical regime and rural development model. Other innovative processes also appear, such as new livestock farming practices and intensive horticulture under greenhouses, which are more discreet and less publicised, but which would play a key role in the construction of new rural development itineraries in the area.

## 4. Materials and Methods

This is qualitative research based on interviews with key informants in the area, carried out in December 2021 and January 2022. In order to identify these key informants, local institutions were consulted, including the INTA (Hilario Ascasubi Experimental center), the Production Department of the Municipality of Villarino, the Producers' Association of Villarino Sur and the Rural Change Programme of the National Ministry of Agriculture, Livestock and Fisheries. The information provided by these institutions made it possible to draw up an initial list of agro-ecological producers, as well as livestock farmers and

traditional producers. The Rural Change Programme's referent provided a list of rural tourism entrepreneurs. Municipal officials such as the Director of Tourism and the Director of Production, a councillor, a technician from the Extension Area of INTA Hilario Ascasubi and a project agent from the Rural Change Programme were also interviewed. With this information, an initial list of 35 candidates were prepared for interview. From all these candidates, only those who met at least two of the following criteria were chosen to be eligible for interviewing:

— Having responsibility in policy design or implementation
— Having responsibility in technical assistance
— Having responsibility in productive associations
— Have been in the area and in the activity for more than 10 years
— Carrying out an innovative activity related to rural tourism
— Carrying out an innovative activity related to agro-ecological production
— Carrying out an innovative activity related to livestock farming

Based on these criteria, 17 actors were identified for in-depth interviews: four livestock and horticultural producers, four tourism service providers, four agro-ecological producers and five technicians and public officials (see Table A1).

The interviews lasted between one and two hours, in a face-to-face format. A guide was used for this purpose, which allowed the following topics to be surveyed (see Appendix B):

— Itineraries of productive development
— Problems for the consolidation of innovative initiatives
— Conflicts or synergies with other stakeholders
— Actors involved
— Mechanisms for the resolution and governance of conflicts and synergies
— Impact of activities on the territory

Once the interviews had been carried out, they were coded and analysed according to the different dimensions of analysis, using qualitative analysis software (Maxqda, 2020). In addition, public documents or other secondary information provided by local and national technical institutions were analysed to better understand the contextual conditions of the innovative processes underway.

## 5. Results

### 5.1. Agroecology at Villarino

Agroecological production in the District of Villarino involves at least 20 producers, all of them very small, with plots of one ha, and in some cases with more than 10 ha, of which only a portion is used for such production. Most of these farms produce a wide variety of vegetables, fruit, honey, small animals (chickens and hens), eggs, and processed foods (jams, preserves). Production practices are carried out without external inputs of agrochemicals, but with manual practices, using natural processes that ensure the total safety of the products. The marketing of these products is done locally, in their own production units, in local village markets, through digital social networks (facebook, whatsapp, instagram), and also with boxes that are prepared once a week with a variety of products, organised by INTA. The creation of the Consejo de Garantías Participativas of Villarino (with the participation of the Universidad Nacional del Sur, agro-technical schools and the Servicio Nacional de Sanidad y Calidad Agroalimentaria), and the issuing of a certificate of local agro-ecological production has improved the marketing of these products to a certain extent.

The factors that have driven these initiatives are linked to three situations:

— Having a rural family origin and the will to continue the productive legacy of grandparents and parents, of healthy products from the garden for family consumption, and currently for sale in nearby towns, "My parents bought this small piece of land and we continue to work what they taught us. We continue to produce as they have done, this gave us great results" (CA, 2021).

— The rejection of conventional forms of production in terms of excessive use of agro-chemicals and consequent deterioration of the environment, and the conviction that agroecology is also a way of life and a new type of relationship with nature [50], "At one point I didn't like the way we were producing vegetables any more. I changed when I saw the amount of agrochemicals we were using in food production" (JP, 2021).

— The discovery of this activity and its recognition as a contribution to income generation, better health and quality of life. This conviction has been made possible through talks, seminars or meetings with technicians or other producers.

The emergence of this type of initiative is not an isolated event in the region. In peri-urban or rural areas close to important localities, there has been an expansion of these groups of agroecological producers, many of them articulated through the Red Nacional de Municipios Agroecológicos (RENAMA), with the support of local universities, or more recently by social movements of urban origin (Movimiento Evita).

These producers have generally gone through three main phases:

1. A phase of discovery and learning about the activity, with few initial products and activities, but incorporating more and more land and production. The key in this stage has been the collective work to be able to train themselves, and land management to be able to produce, since many of them have to plan the use of water, their plots, and eventually some means of tillage, since in many cases they do not have mechanised tools. The role of INTA, and especially of the PROHUERTA programme, has been fundamental at this stage, as it has helped to raise awareness and train producers in these new practices, "they trained me, I learnt and began to incorporate a different vocabulary, and they also introduced me to many experiences of working in vegetable gardens in other places" (CA, 2021).

2. A consolidation phase. At this stage, producers consolidate their production and are even encouraged to diversify with the incorporation of other horticultural crops and even other activities, such as the production of broilers and laying hens in particular. Marketing mechanisms are organised, increasing the scale of production and developing new innovations around new products, and also pest and weed eradication and the use of organic fertilisers, among others. Collective action is key at this stage, both for marketing and training and for the structuring of their networks and recognition mechanisms (stamps, permits, certifications).

3. A stabilisation phase. This last stage becomes a process of reorganisation of activities and products. There is a process of individual and collective learning that leads many of these producers to structure their activities definitively and remain in the same, no longer trying to incorporate new activities or products, but rather to find solutions to any new structural problems that emerge, especially with regard to resource management (e.g., water, pests), and to the forms of marketing that they maintain. In many other cases, learning and experience have led them to reduce their activities and to concentrate on more strategic actions that ensure greater efficiency and profitability of their actions without neglecting agroecological practices in general.

This whole evolutionary process has been immersed in a large number of problems and conflicts, which in one way or another have acted as a brake or constraint on the development of activities. A first problem has been the lack of knowledge about the activities and products, so many producers had to learn to produce under a different model, to work the land, sow and look after their crops in a different way, without depending on chemical inputs. This task has not been easy, as in many cases these smaller producers lacked basic tools for tilling the land. Climate change, water availability and management have also been key factors in the evolutionary process, as permanent droughts and drastic changes in the temperature and wind have caused new conditions that are very difficult to manage for many of these producers who are without previous experience in these activities. A third key factor has been the lack of public support to improve regulations, and to encourage and promote this new way of producing. In many cases, public actors have a much more declamatory presence, but are much less effective in their actions, such

as the local municipality, as there are no global planning strategies to guide the future of this activity [48].

However, perhaps the most important problem has been the lack of local social recognition; there is no appreciation of this productive practice by other producers or the general population of the effort made and the value of the products generated. Although in many cases the local population buys the products produced by these producers, sales remain stable over time thanks to direct sale mechanisms or in boxes. On the other hand, there is greater recognition on the part of a sector of the population in nearby cities, such as Bahía Blanca (300,000 inhabitants), who see this type of enterprise as very important and are more interested in acquiring agroecological products through the internet or in other ways. This lack of recognition or interest at the local level could be linked to three facts: the higher cost of these products compared to conventional horticultural products, the low visibility of these producers and their products, and the lack of trust in these products due to the lack of certifications or regulations.

Another emerging conflict in the development of new agroecological initiatives has been the involvement of political movements, whose main objective is not agroecology, but which support and transfer subsidies and resources to their associates and impose rules of operation or control the participation of producers without the endorsement or participation of the producers as a whole.

The resolution of all these problems and conflicts is attempted in two ways:

- On the one hand, through collective action, i.e., the organisation of a network of agroecological producers. The functioning of these networks has been very effective, as they meet and communicate systematically to discuss the different problems and possible solutions. This has helped to organise and sustain the multiple initiatives [49]. This implies a great effort of participation, which is not always easy due to the great distances between the producers and the working groups.
- On the other hand, the support of INTA, which has been key in the promotion and monitoring of these activities, in the provision of certain resources, training and the organisation of marketing. The participation of this institution is also very important as a reference of technical and institutional quality; the institution's prestige provides a guarantee for the agro-ecological innovation process.

Currently, although efforts have been made to improve and diversify production, the number of producers and the volume of production has not increased significantly, nor has the local demand for these products, so it can be said that it is still a niche activity that has not been able to overcome the scale of producers or products.

### 5.2. Rural Tourism

The District of Villarino has a great diversity of landscape and cultural resources that have been valued at different historical moments, although religious tourism clearly stands out with the presence of Fortín Mercedes, a religious site that summons travellers and tourists from all over the country. Fortín Mercedes is not the only one, there are also the Termas Ceferino Namuncurá (thermal baths) and Lago Parque La Salada that make up the Tourist Triangle of Villarino, that presents the greatest tourist development in the district, as it concentrates the greatest offer of services and attractions. The rest of the district has a variety of attractions that are slowly being developed by different local and regional actors. The great majority of them are organised in two large groups of service providers with at least ten producers in each, characterised by the diversity of products and services they offer, although on a purely family scale.

The predominant activities currently developed are extra-hotel accommodation on a small scale (between two and four cabins), with self-catering or in a small restaurant created in the rural environment, craft production for sale to tourists, and the organisation of events and guided tours of natural areas or cultural heritage resources. In recent years, horseback rides have also been organised, attracting visitors from all over the region, which implies an important organisational and logistical effort that has an economic impact.

As in the agro-ecological activities, these service providers have gone through different evolutionary stages:

1. A phase of learning and organisation of their enterprises. At this stage, these service providers have begun to understand how the market works, the demands of the public, and their own capacities and opportunities for developing the activity. The advice provided by technical advisors, ongoing training, together with visits to other associative enterprises in the region have been key at this stage, "associativism has allowed us to grow and develop. Meeting other people, learning about other people's experiences and learning from the promoter opened our minds and allowed us to create a space for thinking, reflecting and making proposals" (AG, 2021).

2. In a later phase of development of the activity, infrastructure and equipment have been created and consolidated (restaurants, new cabins, tasting rooms, solar panels and heaters, road repairs, signage), which has allowed the activity to consolidate and stabilise. A key factor has again been, firstly, the strengthening of the networks of rural tourism entrepreneurs, which has allowed them to establish better links with the municipality and other bodies (INTA, national government), and secondly, visits to other enterprises, with the consequent benefit of creating new ideas and activities.

3. In a third phase, many of these activities reached a stage of stabilisation, with a defined market and a certain scale, which is often not easy to increase as they are family enterprises.

This evolutionary process was not without problems and conflicts. A recurrent element has been the lack of infrastructure and basic equipment that would allow the development of the activities, especially the lack of road maintenance and signage, and also water and electricity in many areas, "the water problem is the most important. The provision of electricity and the improvement of roads is also something that needs to be strengthened" (CR, 2021). Another recurrent problem is the lack of trained human resources or specialised services necessary for the operation of the enterprises (from personnel for general infrastructure maintenance to personnel for customer service), "the most important obstacle we have now is to get people to do cleaning and maintenance work" (AG, 2021). However, the problem that stands out most is the lack of clear, well-defined and planned public policies with continuity on the part of the municipality and other public bodies. A key indicator of this situation is the tourism. The weak support for the activity has also resulted in significant delays in all the bureaucratic and administrative processes, in the granting of permits and very low level of budget allocated by the municipality authorisation in general, hindering the development of initiatives.

In order to tackle all these problems, the entrepreneurs have strengthened collective action, consolidating their associative groups, organising activities and events jointly and, above all, coordinating with other actors as a group in order to generate greater pressure on the municipal authorities and other bodies. In this sense, there is a strong organisational capacity to articulate and mobilise actors at different levels (national, provincial and municipal), actions that are developed with a great deal of will and without resources on the part of the entrepreneurs themselves. A key role in this process of strengthening the groups of entrepreneurs has been played by the presence of technical advisors and promoters who, through the Cambio Rural Programme, which depends on the Ministerio Agricultura, Ganadería y Pesca de la Nación, accompany the associative work, together with the support of INTA. Both institutions supported the creation, participation and consolidation of these groups, which allowed the entrepreneurs to articulate with each other, develop shared projects, receive training and visit new enterprises in other parts of the country, in order to learn and create new ideas and projects. The involvement of the technicians also allowed the tourism groups to create wider exchange networks at regional and national level, as spaces for learning and creating new opportunities, "the "Rural Change" programme has strengthened collective work. Now the producer looks at his activity, but also at what is happening in his territory, in his community. When they ask for the road to be repaired, the

signs, the electricity, they do it for everyone. Associative work also transcends the group level and has an impact on villages and wider territories" (JC, 2121).

The activities carried out by these tourism service providers have been integrated into the traditional productive fabric of the territory, creating new business and employment opportunities, enhancing the natural and cultural heritage, and reinforcing local identity, while at the same time contributing to building and presenting a different image of the territory and its resources. This effort has allowed the development of an activity that is currently stabilised, which can be considered as a niche activity, well organised and with several support networks, but which cannot increase or overcome the current conditions due to several factors, such as the lack of new actors willing to get involved in new rural tourism activities which would increase the number of actors and activities, the persistence of structural problems (lack of infrastructure, regulatory framework, incentives, etc.), and above all weak public action at the local or provincial level that does not generate proactive structural policies for the development of this activity in the territory.

*5.3. Livestock Farming*

Livestock farming in the District of Villarino is not new. There are hundreds of farms in both the rainfed and irrigated areas that have been engaged in this activity for many decades, most of the time using very traditional methods, with low technology, low input use and medium or low levels of efficiency. However, there are differences between the rain-fed areas, dependent on rainfall, and the irrigated areas, where better pastures are available.

Livestock production has been an activity that is complemented with cereal and oilseed production in most of the farms, and on a few occasions with horticulture, since producers often lease a fraction of their farms for onion production, depending on the climate and soil conditions. The typical production is of beef calves that are raised in the area and then taken to other areas with better pastures for fattening, or cows for beef production.

However, in recent decades, the persistence of dry cycles, soil deterioration, the loss of profitability of intensive activities (especially horticulture) and climate change in general, have prompted many producers to abandon horticultural activities (and stop leasing their land), and intensive cropping that are highly demanding of inputs and water resources, to adopt new production practices in pasture production, for livestock production (for milk or meat), that is, in activities that have always been carried out but under different conditions and with much more innovative management systems. The new practices promoted by producers are characterised by a much more intensive use of high-value inputs (vaccines, genetics, strategic feed for early weaning), the incorporation of more adapted infrastructure for better herd management, and a more intensive use of technological knowledge on livestock management, with a more focused approach to production and environmental sustainability (more efficient use of water, more planned and rational management of fields with the incorporation of shelter belts, integrated management of perennial pastures, and recovery of soil and native forests), "we try to produce in an environmentally sustainable way. We have been managing the field with direct sowing to avoid soil loss due to wind erosion for more than 10 years. This year we are trying to make a lot of forest curtains" (SU, 2022). This process of innovation clearly moves away from the more conventional and traditional practices of lower efficiency, and from the practices in vogue in recent years focused solely on increasing productivity but with a strong environmental impact (feed lots), "I use less and less agrochemicals, I value the cost of the application and the damage. If it is not economically profitable, I don't make the applications and I do a lot of mechanical weed and pasture control, using less and less insecticides and agrochemicals in general" (AF, 2022).

A key element that clearly differentiates these processes of innovation and change from rural tourism and agroecology is that they are led by private action. There are no links with public action, and the link with collective action is on a technical level, with the novel presence in the area of producer organisations such as AAPRESID (Asociación Argentina

de Productores en Siembra Directa) and CREA (Consorcios Regionales de Experimentacion Agricola). Organisations linked to the administration of resources (e.g., vaccination) are also key, especially water resources, such as the Consorcio Hidraulico.

These processes of innovation and development of new activities have not had to face certain key challenges, such as produce marketing, as these systems were historically organised and structured by the markets themselves. In this sense, this situation constitutes a great advantage in comparison to rural tourism and agroecology, activities in which marketing is a dimension that has to be developed.

Clearly, these processes of innovation and change in livestock farming are integrated into the productive fabric of the territory, creating new activities and more complex services with greater incorporation of knowledge (technical assistance services, sale of inputs).

Finally, and perhaps the most important element in this process of change, is that the producers in general see these innovations as an alternative way to build a new socio-technical regime, i.e., they are betting on an activity in which there is already a certain tradition and knowledge, and around which there are already structured marketing systems. It could be said that this is an innovation process conditioned by existing structures.

Table 1 presents a synthesis of the three innovation processes. It allows these processes to be compared on the basis of their main variables.

**Table 1.** Summary of the main characteristics of three innovations processes.

| Variables | Agroecology | Tourism | Livestock |
|---|---|---|---|
| Production development pathways | Initial growth, now stabilized | Initial growth, now stabilized | Permanent growth |
| Main limitations | Infrastructure problems (water, electricity), lack of equipment, lack of financing and lack of support for promotion | Infrastructure problems (roads, energy, water), permits, lack of support for promotion | Infrastructure problems (water) |
| Commercialisation | Commercialisation of local sales, low structuring | Direct sales to regional audiences | Structured sales by consolidated markets |
| Social recognition | Low or none | Medium | High |
| Synergies maintained with other actors | None or very low | High | High |
| Actors | Reconverted traditional producers, neo-rural | Traditional settlers, women | Traditional producers |
| Problem-solving arenas | INTA | INTA | Producers' organisations, private action |
| Policy dialogue, governance | Collective action, strong dependence on public action (INTA). | Collective action articulated with public action (INTA) and regional networks | Private action, with collective action, low dependence on public action |
| Impact of activities on the territory | Null or very low | Low | Medium or high |

## 6. Discussion

### 6.1. The Framework Provides an Understanding of the Processes of Innovation and Rural Change

**The MLP leads to understanding the innovation process in the territory of analysis.** The analysis following the MLP approach allows us to interpret the innovation process underway in the District of Villarino. The cases of agro-ecology and rural tourism analysis allow us to observe how these initiatives constitute innovative activities [13–15,39], with a strong specificity that depends on local conditions, and which end up consolidating themselves as niche activities. The theory postulates that, over time, these activities become integrated into the traditional production systems of each area, generating new socio-

technical regimes [17]. This situation can be perceived for the moment in the local case, although with profound differences according to the type of activities analysed.

**Agroecological initiatives follow a logic of co-presence, which makes scaling up difficult**. Within this process of innovation and change, agroecological initiatives follow a logic of co-presence, i.e., activities and actors are deployed in the same area but do not coexist or articulate with other local activities [18], there is a strong lack of awareness among actors and activities, either due to a lack of recognition, or because of their low economic weight and very small workforce [24]. They have difficulties in marketing their products [50], and maintain a very low level of linkage with the dominant socio-technical regime (intensive horticulture and traditional crop and livestock farming), which does not allow for synergies or joint development projects [25]. Despite this, these initiatives have received strong technical and organisational support from INTA, the Universidad Nacional del Sur and other networks of projects and organisations, and have built a logic of collective action that has allowed them to become self-supporting over time, although without being able to make significant quantitative leaps.

**Rural tourism initiatives follow a logic of co-existence that allows for their consolidation**. These activities are visible, recognised and accepted by multiple local actors, as they are integrated into the productive matrix and consumed by the local population or other external actors, which at a certain point could be a key element for the construction of a more synergetic and comprehensive territorial project [21]. On the other hand, there are shared incentives for these activities to develop, the local population benefits from the dynamics of the construction of tourist services (improved roads, electricity, water supply, rural internet, etc.), from the protection of the landscape, and from the processes of activation of historical-architectural resources.

The processes of innovation in livestock farming follow a logic of co-existence and point the way towards a new socio-technical regime. The emerging processes with the greatest capacity to build a new socio-technical regime are not rural tourism and agroecology, but the new forms of livestock production, i.e., activities that are based on what already has a certain tradition and organisation in the area (path dependency) [33], and knowledge on the part of producers, and for which structured and organised markets already exist. Following Geels, we could predict that in the coming decades the new livestock activities will go from being niche activities to structuring the functioning of the local socio-technical regime, since there are marked conditions for the scaling up of these activities.

*6.2. There Are Factors That Limit or Enhance the Process of Innovation and Socio-Technical Regime Change*

The three experiences analysed allow a comparative reflection on the key factors that limit or, on the contrary, enhance innovation processes and allow the passage from niche activities to the construction of new socio-technical regimes. Several factors can be observed:

Identity. Innovative processes and activities must have their own identity and their own discourse on the way they are organised and their development itineraries [18]. This discourse and identity can be constructed by the producers themselves or by the technical support organisations, but it is important that this message is clear and allows these innovations to be made known in the same territory and with other actors at different multi-scale levels. This identity and discourse allow the actors to affirm their activities and projects at the local level, but also, and as a very important achievement, to insert themselves into networks of productive projects or organisations that go beyond the local level (Red Nacional de Municipios Agroecológicos, Red de Turismo Rural, AAPRESID, etc.). This case study shows that the three innovation processes have been able to create their own identity and a technical discourse that supports them and allows them to position and justify themselves in the local context and in the eyes of other actors [22].

**Social recognition.** The innovation processes with the best chances of moving from niche situations to structuring new production regimes are those that are recognised and accepted by other actors as a vital part of the territory [18]. This recognition translates concretely into integration in local economic life, through the purchase of their products, the provision of services, and active participation in the construction of local collective actions. This recognition has been very clear in the case of livestock farming, less so in the case of rural tourism and nil in the case of agroecology, which limits the possibilities of moving from being niche activities to structuring new socio-technical behaviours or regimes, at least in the short term.

**Support policies**. Research has shown that public support has been very uneven. Agroecological initiatives and rural tourism have been supported by different bodies and institutions, such as the Ministerio de Agricultura, Ministerio de Desarrollo Social and INTA, who have been involved in promotion, organisation of producers, organisation of resources, training and technology transfer, and to a much lesser extent by the local municipality. However, the low level of coordination [41], and the weakness and discontinuity of support over time has not resulted in these initiatives to be scaled up. Innovation processes in livestock farming have had no public support, which has been replaced by the collective support and private action. The results show that it is not the presence of public promotion policies that guarantee the development and scaling up of innovation processes; on the contrary, the initiatives that emerge with the greatest potential for socio-technical regime change are those that do not have such support and are based on private and collective action.

**Knowledge building**. The results show that innovation processes require the construction of new knowledge, which can be used for the consolidation and scaling up of innovative experiences. Nevertheless, the research shows clear differences in the types of information generated and shared [14,51]. Local or tacit knowledge is prevalent in agroecological and rural tourism initiatives, including knowledge, skills and competencies that are created and reproduced through varied and complex forms of social interaction [52]. They are often the result of particular ways of doing things, which are transmitted informally and reproduced by the actors in their usual practices in a singular way, hence their original character and their difficult replicability and transmission. The construction of this knowledge and its exchange is supported by public agencies (INTA) and by networks of producers who exchange knowledge on the basis of local experience. In livestock innovations, on the other hand, although local knowledge is constructed, codified knowledge predominates, the latter is produced in the sphere of scientific activity and exchanged through technical language, formal education and manuals; it is proven to be applicable in multiple contexts and reproduced in a univocal manner, while it is possible to acquire it through different market channels. In this case, knowledge is much more widely disseminated by producers' organisations themselves and by national and international laboratories (private and collective action).

**Effective marketing mechanisms**. Activities with the potential to overcome niche instances are those that have developed effective and sustained marketing mechanisms over time, or that are built on the basis of marketing systems that are already organised. In the case of agroecology and rural tourism, the mechanisms are not clearly structured and are very random; they are mechanisms in the process of being created, which generates many difficulties and a high degree of uncertainty about their development. On the other hand, in livestock farming, marketing mechanisms have been in place for decades, which secures the activity and allows for adequate planning of production.

**Governance**. Adequate organisation of actors and institutions, clear rules of the game and trust are fundamental for the construction of innovation processes [43]. On the contrary, the higher the levels of arbitrariness, permanent changes in the rules of the game and lack of institutional organisation, the lower the capacity to build virtuous innovation dynamics. The cases analysed show how changes in the rules of the game, or the lack of a more structured or institutionalised collective organisation, inhibit the

innovative process (agroecology), while actions that are developed in a context of more structured governance, based on historical knowledge among the actors, allow for better consolidation of innovative processes [40]. This suggests that the construction of clear rules of the game within organisations, compliance with them and trust between actors are key to the sustainability of innovation processes. Research has also highlighted the importance of collective action and networks in the construction of innovative processes, especially in non-traditional activities that are developed without any previous experience. It is through these collective actions that actors can articulate, agree on initiatives, and build learning dynamics. On the other hand, the presence of multiscale networks is a very important resource as it allows the articulation of numerous actors from different levels of political and administrative organisation (Nation, Province, Municipalities), in order to obtain resources that cannot be obtained at the local level [53]. The capacity to build collective action and multi-scalar networks has been remarkable in the case of agroecology and rural tourism, which has allowed them to be self-supporting despite the severe constraints inherent to the activities.

**Development path dependency is central to the innovation process.** Clearly, the weight of territorial conditions, productive tradition, local culture and market conditions are structuring the innovation processes [54]. Innovative processes that are based on activities already present in the territory and that involve processes of readaptation or incremental improvements have enormous possibilities of scaling up their innovation processes and consolidating new socio-technical regimes, as is the case of livestock farming in the territory under analysis. Moreover, these innovative processes have a much greater capacity to overcome the structural problems imposed by the territory, such as the lack of water, roads, electricity, changing climatic conditions, the lack of productive equipment and local and national regulatory norms. In the case under analysis, livestock activities have been the most capable of overcoming these structural limitations, not only because they have greater economic resources, but also because they have had more structured resource governance models over time. Thus, agroecology and rural tourism are seen in the territory as "newcomers".

## 7. Conclusions

This article analyses the capacity of different types of innovative processes to consolidate and scale up, with a view to transforming the socio-technical regime of a rural territory. The qualitative and exploratory research focused on the analysis of the trajectory of agroecology, rural tourism and livestock farming through key informants.

We used the conceptual framework inspired by Geels, the multilevel perspective (MLP), which we consider to be a very useful heuristic tool for understanding innovation processes. However, we also incorporated the concepts of co-presence and co-existence of activities, as they allowed us to observe the level of integration of the innovation processes in current socio-technical regimes, which also allowed us to understand the level of relationship of the activities with their territorial context, and with other activities and actors. We also incorporated the concept of governance, as it provided us with tools to understand the ways in which innovation processes are linked and organised in the evolutionary processes of rural territories.

Based on the analysis of innovation processes in agroecology, rural tourism and livestock farming, three key ideas can be concluded.

The passage from niche situations to new socio-technical regimes would be determined by several factors, but the level of recognition and acceptance that an activity has in the area [18] and path dependency stand out clearly [33]. Without recognition by the actors in the territory, there is no possibility of building virtuous articulations that would allow the integration and scaling up of these niche actions. On the other hand, the structural conditions, history and tradition of productive activities are very important, as there is a cultural inertia in carrying out these activities; activities that have a local history are more likely to be transformed and generate new innovations that are accepted in the local

context than other types of new activity. As an example, and following Anderson in his analysis of agro-ecology [22], in order to scale up innovative processes, and move from a logic of co-presence to co-existence, a much broader territorial base must be created, a larger scale of cases, with greater connections in the same area, in order to break out of isolation. Networks and multi-scale articulations, which have been created in the case of agroecology and tourism, would not be enough; a larger local base is needed.

Secondly, the model of technical assistance and governance of rural development processes based primarily on public action does not seem to be as successful as in other historical moments. Although many efforts have been made to promote agroecology and rural tourism, it seems that this is not enough; there is a lack of articulation with private action. Collective and public action would be acting as a factor in sustaining innovation processes in the territory but would not contribute to the scaling up of innovative processes or to the transition from a scenario of co-presence to co-existence.

Thirdly, and from a conceptual and methodological point of view, the usefulness of the concepts of co-presence and co-existence as tools for understanding how transition processes can be constructed has been verified. These concepts complement the MLP proposals in a very effective way, as they allowed us to clearly identify the factors that limit or impede the passage from niche actions to the construction of new socio-technical regimes. Clearly this has a high value in terms of the design of policies and strategies for rural transition and development.

In view of these findings, we consider that there are two main research paths regarding innovation and transition in rural areas of Argentina.

A first research path should place more emphasis on not only analysing new products or niche activities (product innovations), but also new technological processes or process innovations based on products that are already traditional in the area, as in the specific case of livestock. In other words, analysing innovation around the activities that have always structured or organised the rural territory, based on the fact that the weight of territorial structures, tradition and existing markets are very important as they again condition emerging processes.

A second key issue for analysis is the role of private and public action in the emergence of these processes of transition and construction of new socio-technical regimes. It seems that public institutions (INTA, universities, provincial and national organisations) are much more focused on promoting innovation and the creation of new activities and the diversification of productive processes, but this would not result in them overcoming niche products and building new socio-technical regimes. Private action would be much more focused on the promotion of processes based on path dependency, as in the case of livestock farming, and would have a greater capacity to contribute to the transition from niche scenarios to new socio-technical regimes.

These new questions could well appeal to the conceptual framework raised in this paper, which have shown their usefulness in advancing the interpretation of the processes of change and the construction of new models of rural development.

**Author Contributions:** Conceptualization, M.S.; methodology, M.S.; validation, M.S.; formal analysis, M.S.; investigation, M.I.H. and M.B.N.; resources, M.S.; writing—original draft preparation, M.S.; writing—review and editing, M.S.; project administration, M.S.; funding acquisition, M.S. All authors have read and agreed to the published version of the manuscript.

**Funding:** This research was funded by the EARTH project, number 598839-EPP-1-2018-1-IT-EPPKA2-CBHE-JP, financed by the ERASMUS Programme of the European Union (Education, Audiovisual and Culture Executive Agency (EACEA). European Union).

**Institutional Review Board Statement:** The study was conducted according to the guidelines of the National Law 25.326 on the Protection of Personal Data of Argentina.

**Informed Consent Statement:** Informed consent was obtained from all subjects involved in the study.

**Data Availability Statement:** Not applicable.

**Acknowledgments:** To the entrepreneurs, technicians and other actors involved in rural development in the District of Villarino for the information provided during the field interviews. To Mariano Pla for reading the first version of the document and his important recommendations.

**Conflicts of Interest:** The authors declare no conflict of interest.

## Appendix A

**Table A1.** Description of interviewed.

| Activities | Name | Place | Date of Interview |
|---|---|---|---|
| Agroecology | JC | Pedro Luro | 23 December 2021 |
| | CA | Hilario Ascasubi | 3 December 2021 |
| | AF | Pedro Luro | 30 December 2021 |
| | AP | Medanos | 3 January 2022 |
| Tourism | AG | Pedro Luro | 3 January 2022 |
| | CR | La Chiquita | 23 December 2021 |
| | EC | Argerich | 23 December 2021 |
| | ML | Argerich | 24 December 2021 |
| Livestock | SU | San Adolfo | 29 December 2021 |
| | AF | Pedro Luro | 3 December 2021 |
| | MP | Pedro Luro | 29 December 2021 |
| | PU | Mayor Buratovich | 4 January 2022 |
| Technicians and others | LR | Medanos | 29 December 2021 |
| | MK | Medanos | 23 December 2021 |
| | BR | Pedro Luro | 3 January 2022 |
| | PP | Hilario Ascasubi | 22 December 2021 |
| | JC | Villarino | 27 December 2021 |

## Appendix B. Survey Questions

**Interview guide for tourism, agro-ecology, and livestock entrepreneurs**

1. What was the origin and the different stages that this enterprise or activity went through? What objectives did they set themselves?
2. What were the main problems you encountered in carrying out this initiative?
3. What were the main conflicts you had to carry out in this experience? (criticism, ignorance, accusations, open conflicts, etc.) With whom did you have these conflicts?
4. Was there or is there an environment or institution where these conflicts could be raised and discussed and eventually resolved? What is this environment and how did it work? What role did the state play in the management or resolution of these conflicts?
5. Do you think that there are people capable of leading and promoting meetings and dialogue between the different local productive sectors? How did these actors act?
6. What were the main synergies or links generated with other sectors? With whom?
7. Do you consider that this activity is still a niche activity (alone in the area) or are there already several and are they consolidated and accepted in the area?
8. What impact does this innovation have on the territory and do these innovations contribute to change the form of organisation and development of the area?

**Interview guide for technicians and public officials**

1. New productive activities have appeared in the area, such as rural tourism, agroecology, etc. Do you know these activities and their producers?

2.   Do you know how they started and how did their activity go?

3.   Were any links or joint projects generated with these new activities in the area? With whom?

4.   Did you have any kind of conflict or different point of view with them?

5.   In case there was any kind of conflict, was there or is there any place or institution where these conflicts can be raised and discussed and eventually resolved? What is this place and how did it work?

6.   Are there actors capable of leading and promoting the meeting and the construction of joint projects with these new ventures? How did these work?

7.   Do you consider that these activities are still special cases, or are they already consolidated and accepted in the area?

8.   What do you think is the impact of these innovative processes in the area, and do these innovations contribute to change the way of organisation and development of the area?

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
