# Peer review of "Constructing the Transitions and Co-Existence of Rural Development Models"

_sustainability, doi:10.3390/su14084625_

Round 1
Reviewer 1 Report
The paper “Constructing the Transitions and co-Existence of Rural Development Models” provides a detailed picture of new experiences undergoing in Argentina, showing how the process of transition to different rural development models is constrained by both the landscape and the dominant socio-technical regime.
The paper provides a clear introduction to the issue although a better clarification of its aims could improve its readability.
The authors provide an overview of their methodology in section 4. Nevertheless, it is quite hard to understand how and where the different input provided by the interviews are included into results and discussion sections. From my side, I think this is the main weak point of the paper. Authors are encouraged to show how they took in account the interviews they made.
Please uniform the language in fig. 1
Author Response
|
Evaluator 1 |
Responses or corrections |
|
The paper provides a clear introduction to the issue although a better clarification of its aims could improve its readability. |
Corrected |
|
It is quite hard to understand how and where the different input provided by the interviews are included into results and discussion sections. Authors are encouraged to show how they took in account the interviews they made |
We included the methodological strategy to determine the actors to be interviewed. We included the sample criteria. Thus, 17 actors were selected to be interviewed. We included textual quotations from the interviewees that give an account of the innovation processes. |
|
Please uniform the language in fig. 1 |
Corrected |
Reviewer 2 Report
This article analyses the innovative processes of agro-ecology, rural tourism and livestock farming in Argentina based on the conceptual framework inspired by Geels, the multilevel perspective (MLP). This work fits well the aims and scope of Sustainability and provides good understanding of the development in rural Latin America.
However, Sustainability is an international journal and it is important for the author(s) to explain why this research is of interest to international readers.
Author Response
|
Evaluator 2 |
Responses or corrections |
|
Sustainability is an international journal and it is important for the author(s) to explain why this research is of interest to international readers. |
In recent decades, Argentina has had a strong agro-export development, thanks to a highly mechanised agriculture, with high levels of use of chemical inputs, which has generated a significant environmental impact in recent decades. The transition process that is being observed is very interesting because of the potential conflicts it may generate with the current productive logic. Understanding these dynamics is extremely useful in order to compare them with other similar dynamics in countries with a strong tradition of export production, such as Latin American countries and, in the near future, African countries, as well as Australia, the United States and other countries with an agro-exporting vocation. |
Reviewer 3 Report
The paper is a qualitative analysis: this kind of researches require a high level of accuracy in order to be considered adequacy. However, the paper is lacking about the practical methodological organization of the research. In M&M section there a very lacking description of the survey applied, therefore information provided into the results section are not clear to be understood. Please, provide a better description of the questionnaire used or the questionnaire itself in the additional material. Moreover, the definition of the sampling is lacking. How the long list has been created before the application of the criteria? How the final stakeholders cover such a criteria? Did all of them cover all the three experiences or just some of them?
Some additional remarks:
- please make a professional proofreading of the documents, there are many typos; Figure 1 and 4 contained some Spanish phrases; at ln 242 a new paragraph begin with However.
- in ln 153- 180 there is a description of the Geels' multi-level model, however there are no references at all.
-
Table 1: few cases or many cases is a too approximated measurement, please provide some quantification
Author Response
|
Evaluator 3 |
Responses or corrections |
|
However, the paper is lacking about the practical methodological organization of the research. |
A detailed explanation is given of how the field work progressed, first by contacting key informants to determine the sample of farmers to be interviewed. The topics that are addressed in the interview are included, and the number of actors that are interviewed is explained (Annex A). |
|
Please, provide a better description of the questionnaire used or the questionnaire itself in the additional material. |
The in-depth interview guide questionnaire is included in annex B |
|
The definition of the sampling is lacking. How the long list has been created before the application of the criteria? How the final stakeholders cover such a criteria? Did all of them cover all the three experiences or just some of them? |
The way in which the sample of producers to be interviewed was prepared is detailed. The factors that were taken into consideration for their selection are detailed. |
|
Figure 1 and 4 contained some Spanish phrases; at ln 242 a new paragraph begin with However. |
Corrected |
|
ln 153- 180 there is a description of the Geels' multi-level model, however there are no references at all. |
Corrected |
|
Table 1: few cases or many cases is a too approximated measurement, please provide some quantification |
Corrected |
Round 2
Reviewer 1 Report
Dear authors, all the raised issues have been addressed. The paper has been substantially improved. Good luck!
Reviewer 3 Report
Although the authors made not so many changes, the quality of the paper raised up significantly. The paper can be published in the present form